# The primary structural photoresponse of phytochrome proteins captured by a femtosecond X-ray laser

Elin Claesson[1†], Weixiao Yuan Wahlgren[1†], Heikki Takala[2,3†], Suraj Pandey[4], Leticia Castillon[1], Valentyna Kuznetsova[2], Léocadie Henry[1], Matthijs Panman[1], Melissa Carrillo[5], Joachim Kübel[1], Rahul Nanekar[2], Linnéa Isaksson[1], Amke Nimmrich[1], Andrea Cellini[1], Dmitry Morozov[6], Michał Maj[1], Moona Kurttila[2], Robert Bosman[1], Eriko Nango[7,8], Rie Tanaka[7,8], Tomoyuki Tanaka[7,8], Luo Fangjia[7,8], So Iwata[7,8], Shigeki Owada[8,9], Keith Moffat[10], Gerrit Groenhof[6], Emina A Stojković[5], Janne A Ihalainen[2], Marius Schmidt[4*], Sebastian Westenhoff[1*]

[1]Department of Chemistry and Molecular Biology, University of Gothenburg, Gothenburg, Sweden; [2]Department of Biological and Environmental Science, Nanoscience Center, University of Jyvaskyla, Jyvaskyla, Finland; [3]Department of Anatomy, Faculty of Medicine, University of Helsinki, Helsinki, Finland; [4]Physics Department, University of Wisconsin-Milwaukee, Milwaukee, United States; [5]Department of Biology, Northeastern Illinois University, Chicago, United States; [6]Department of Chemistry, Nanoscience Center, University of Jyvaskyla, Jyvaskyla, Finland; [7]Department of Cell Biology, Graduate School of Medicine, Kyoto University, Kyoto, Japan; [8]RIKEN SPring-8 Center, Hyogo, Japan; [9]Japan Synchrotron Radiation Research Institute, Hyogo, Japan; [10]Department of Biochemistry and Molecular Biology and Institute for Biophysical Dynamics, University of Chicago, Chicago, United States

*For correspondence:
smarius@uwm.edu (MS);
sebastian.westenhoff.2@gu.se
(SW)

†These authors contributed
equally to this work

Competing interests: The
authors declare that no
competing interests exist.

Reviewing editor: Werner
Kühlbrandt, Max Planck Institute
of Biophysics, Germany

**Abstract** Phytochrome proteins control the growth, reproduction, and photosynthesis of plants, fungi, and bacteria. Light is detected by a bilin cofactor, but it remains elusive how this leads to activation of the protein through structural changes. We present serial femtosecond X-ray crystallographic data of the chromophore-binding domains of a bacterial phytochrome at delay times of 1 ps and 10 ps after photoexcitation. The data reveal a twist of the D-ring, which leads to partial detachment of the chromophore from the protein. Unexpectedly, the conserved so-called pyrrole water is photodissociated from the chromophore, concomitant with movement of the A-ring and a key signaling aspartate. The changes are wired together by ultrafast backbone and water movements around the chromophore, channeling them into signal transduction towards the output domains. We suggest that the observed collective changes are important for the phytochrome photoresponse, explaining the earliest steps of how plants, fungi and bacteria sense red light.

## Introduction

Phytochrome photosensor proteins are crucial for the optimal development of all vegetation on Earth (*Butler et al., 1959*; *Gan et al., 2014*; *Quail et al., 1995*). Prototypical phytochromes can exist in two photochemical states with differential cellular signaling activity, called red light-absorbing (Pr) and far-red light-absorbing (Pfr) state (*Figure 1—figure supplement 1*). As a result, phytochromes

**eLife digest** Plants adapt to the availability of light throughout their lives because it regulates so many aspects of their growth and reproduction. To detect the level of light, plant cells use proteins called phytochromes, which are also found in some bacteria and fungi. Phytochrome proteins change shape when they are exposed to red light, and this change alters the behaviour of the cell. The red light is absorbed by a molecule known as chromophore, which is connected to a region of the phytochrome called the PHY-tongue. This region undergoes one of the key structural changes that occur when the phytochrome protein absorbs light, turning from a flat sheet into a helix.

Claesson, Wahlgren, Takala et al. studied the structure of a bacterial phytochrome protein almost immediately after shining a very brief flash of red light using a laser. The experiments revealed that the structure of the protein begins to change within a trillionth of a second: specifically, the chromophore twists, which disrupts its attachment to the protein, freeing the protein to change shape. Claesson, Wahlgren, Takala et al. note that this structure is likely a very short-lived intermediate state, which however triggers more changes in the overall shape change of the protein.

One feature of the rearrangement is the disappearance of a particular water molecule. This molecule can be found at the core of many different phytochrome structures and interacts with several parts of the chromophore and the phytochrome protein. It is unclear why the water molecule is lost, but given how quickly this happens after the red light is applied it is likely that this disappearance is an integral part of the reshaping process.

Together these events disrupt the interactions between the chromophore and the PHY-tongue, enabling the PHY-tongue to change shape and alter the structure of the phytochrome protein. Understanding and controlling this process could allow scientists to alter growth patterns in plants, such as crops or weeds.

can distinguish two colors of light, providing plants, fungi, and bacteria with primitive two-color vision. Light is detected by a bilin chromophore, which is covalently linked to the photosensory core of the protein (*Wagner et al., 2005*), comprising of PAS (Per/Arndt/Sim), GAF (cGMP phosphodiesterase/adenyl cyclase/FhlA) and PHY (phytochrome-specific) domains. Two propionate side chains additionally anchor the chromophore non-covalently to the protein (*Figure 1b*). The signaling sites of the phytochrome are found in its C- and N-terminal output domains, which vary between species. Important for the signaling is a stretch of amino acids in the PHY domain, called the PHY-tongue, which changes from a β-sheet in Pr into an α-helix in Pfr state (*Essen et al., 2008*; *Yang et al., 2008*; *Takala et al., 2014*; *Sanchez et al., 2019*) The chromophore connects to the PHY-tongue via a strictly conserved aspartic acid, which is expected to play a crucial role in signal transduction.

Key to phytochrome function is the primary photoresponse on picosecond time scales. Here, light signals are translated into conformational changes. The changes arise in the electronically excited bilin, but must then be transduced to the surrounding protein residues. This prepares the protein for a formation of the first intermediate (Lumi-R for prototypical phytochromes), in which isomerization of the D-ring has likely occurred (*Rüdiger et al., 1983*; *Dasgupta et al., 2009*; *Yang et al., 2012*; *Rockwell et al., 2009*; *Ihalainen et al., 2018*). The mechanism that leads to the first intermediate is currently not well understood, because crystallographic observations of phytochromes directly after photoexcitation have not been available.

## Results

To address this gap of knowledge, we recorded time-resolved serial femtosecond X-ray crystallographic (SFX) data of the PAS-GAF domains of the phytochrome from *Deinococcus radiodurans* (*Dr*BphP$_{CBD}$) at 1 ps and 10 ps after femtosecond optical excitation. The experiments were performed in Japan, using the SPring-8 Angstrom Compact Free Electron Laser (SACLA) tuned to 7 KeV (*Tono et al., 2015*). For homogeneous excitation of the crystals, we photoexcited micrometer-sized crystals in a grease jet with a photon density of 1.7 mJ/mm$^2$ ($1/e^2$ measure, see Materials and methods) into the flank of the absorption peak at 640 nm (*Figure 1—figure supplement 1*). Taking into account the significant light scattering in the grease-buffer mixture (*Figure 1—figure*

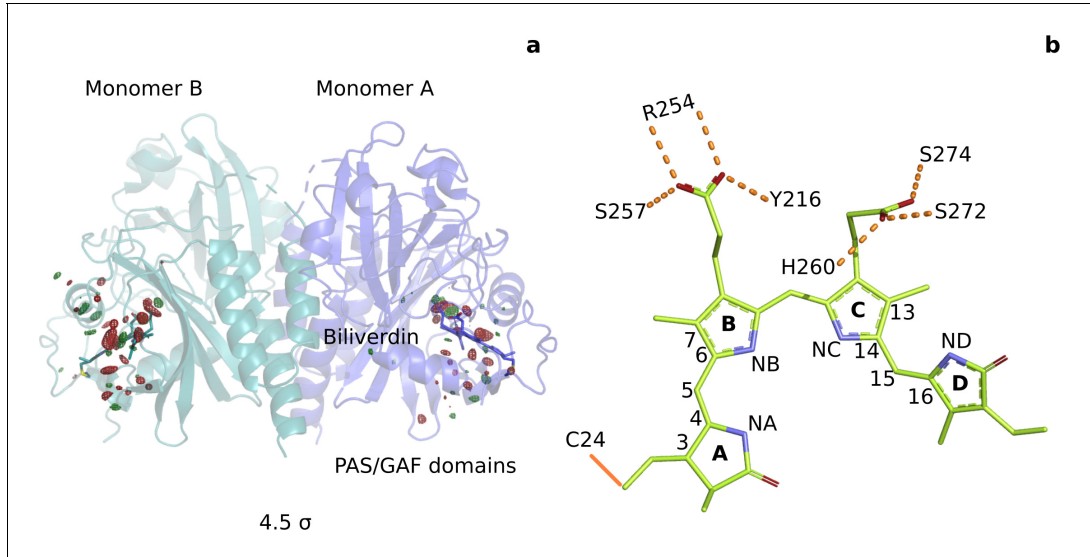

**Figure 1.** Photoinduced observed difference electron density features are focused on the chromophore binding pocket. (**a**) The observed difference electron density map at 1 ps is displayed together with the *Dr*BphP_dark structure. Red and green electron density peaks, contoured at 4.5 σ, denote negative and positive densities, respectively. Monomer A is colored blue and monomer B is in aqua. (**b**) Schematic illustration of the biliverdin chromophore. The hydrogen-bonding networks between the propionate groups and the protein are marked with dashed lines. In *Dr*BphP, the chromophore is covalently linked to a cysteine residue in the PAS domain (solid line).

The online version of this article includes the following figure supplement(s) for figure 1:

**Figure supplement 1.** The photocycle of *Deinococcus radiodurans* phytochrome (*Dr*BphP) and the spectral properties of its PAS-GAF fragment in the microcrystalline form.
**Figure supplement 2.** Microcrystals used for SFX data acquisition in SACLA.
**Figure supplement 3.** Unit cell parameters distribution of SFX data sets.
**Figure supplement 4.** Significant difference electron density features observed.
**Figure supplement 5.** Comparison of the observed difference electron density maps at 1 ps and 10 ps.

*supplement 1*), we estimate that the average number of photons per chromophore is 0.5–1 (see Materials and methods). We recorded the SFX data at 1 ps for several excitation fluences (*Figure 2*). Lowering the excitation density tenfold from 1.7 mJ/mm² photons to 0.2 mJ/mm² resulted in a joint reduction of all difference signals. Critical signals, like the twist of the D-ring and the photodissociation of the pyrrole water from the chromophore sustained when lowering the excitation densities, indicating that the signal arises predominately from one-photon excitation. The refined structure in dark (*Dr*BphP_dark), 2.07 Å resolution (*Table 1*), was very similar to our previous dark structure solved by SFX (5K5B, RMSD 0.646 Å and 0.610 Å for monomers A and B) (*Edlund et al., 2016*), but the present crystals contained two monomers in the asymmetric unit (*Figure 1a*, *Figure 1—figure supplement 2*, *Figure 1—figure supplement 3*). The refined 1 ps structure was solved to 2.21 Å (*Table 1*).

From the time-resolved data, we calculated Fourier difference electron density maps ($|F_o|^{light} - |F_o|^{dark}$), which report on the change of structure due to optical excitation (see Materials and methods). Briefly, the diffraction data for light and dark were scaled to each other and subtracted, assuming preservation of the phases (see Materials and methods for details). The map at 1 ps indicates many significant changes in difference electron density (*Figure 1a*) above the background level of 3.0 standard deviations (σ) (*Figure 1—figure supplement 4*). The changes cluster around the chromophore, with the strongest negative densities for the pyrrole water (monomer A: −8.2σ, B: −9.4σ, *Table 2*). The map at 10 ps contains similar significant features, but at weaker overall intensity (pyrrole water A: −5.0σ, B:−6.5σ) (*Figure 1—figure supplement 4* and *Figure 1—figure supplement 5*). We ascribe this to a lower population of the activated state at 10 ps compared to 1 ps. Monomer A has a lower signal strength than monomer B, but provided a clearer difference map around the chromophore. We refined a structural model (*Dr*BphP_1ps) using extrapolated

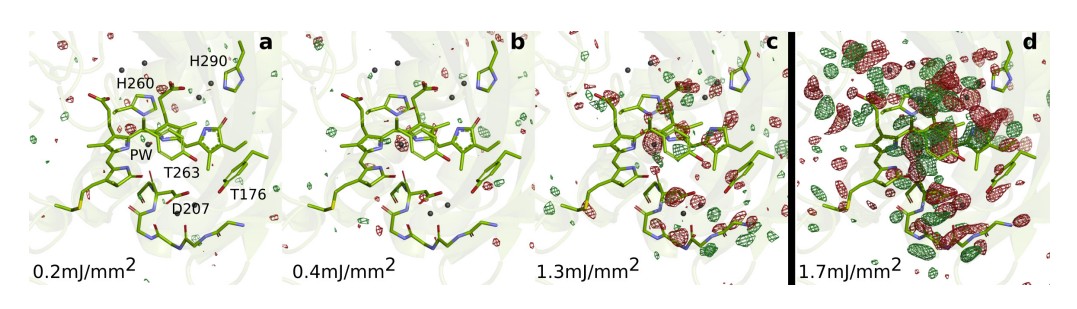

**Figure 2.** SFX data as a function of excitation fluence. The _Dr_BphP$_{dark}$ structure (green) is shown together with the observed difference electron density, contoured at 3.5 σ, at 1 ps collected using (**a**) 0.2 mJ/mm², (**b**) 0.4 mJ/mm², (**c**) 1.3 mJ/mm², and (**d**) 1.7 mJ/mm². All spot sizes were computed assuming Gaussian line shapes with the $(1/e^2)$ convention. The data shown in panel A-C were collected at SACLA in May 2019, whereas the data shown in panel D was collected in October 2018. The same experimental setup was used in both occasions. The laser energy of the experiment in 2018 can be found in the Materials and methods section. The energies for the experiment in 2019 were 16 μJ, 42 μJ, and 106 μJ (panels A-C, respectively). During the experiment in 2019, the femtosecond laser beam was misaligned by 50 μm distance from the interaction spot between X-rays and jet in the direction of flow. The laser intensities were corrected for this displacement assuming a Gaussian line shape. The excitation fluence is similar to previous femtosecond time-resolved SFX experiments (**_Nogly et al., 2018_**; **_Pande et al., 2016_**; **_Barends et al., 2015_**; **_Coquelle et al., 2018_**); however, we found high scattering in the grease/buffer mixture (**_Figure 1—figure supplement 1_**). Since the crystallographic signals were reduced when lowering the excitation fluence and disappeared completely when reaching 1/10 of the maximum value, we conclude that the excitation fluence that actually reaches the crystals in the grease matrix is much lower than the incoming photon fluence and that the photoexcitation is in the single-photon regime.

structure factors (**_Figure 3—figure supplement 1_**; **_Pande et al., 2016_**). The refinement of the structure against the 1 ps data was successful using a photoexcitation density of 8%. However, we aborted our attempts to refine a structural model against the 10 ps data, as the model would have become unreliable due to an even lower photoactivation yield. We focus our discussion on monomer A and the 1 ps time point, although all conclusions are supported by monomer B and the features observed in the difference maps at 10 ps (**_Table 2_**, **_Figure 1—figure supplement 5_**).

First, we inspect the D-ring region at 1 ps (**_Figure 3_**). We observe strong negative difference density features on the atoms of the D-ring (marked I, II, III), correlating with density gains at both faces of the ring (IV, V, VI). These features strongly indicate that the D-ring twists. The positive feature IV homes the N-H and C = O groups, whereas V and VI indicate densities for the methyl and vinyl groups in the twisted ring (**_Figure 3c_**). Excellent agreement was obtained between the observed difference map and the difference map calculated from _Dr_BphP$_{1ps}$ ($F_c^{1ps} - F_c^{dark}$), when the D-ring twists (C14-C15-C16-ND) from around 20° in the dark to 60° monomer A) and 90° monomer B) at 1 ps (**_Figure 3c and e_**). Although the twisting movement is clearly indicated by the difference map, we judge the precision of the angle to be low and approximately ±25°.

Concomitant with the twist of the D-ring, the C-ring translates by approximately 0.69 Å as indicated by the correlated negative (VII) and positive (VIII) electron density (**_Figure 3a_**). Furthermore, the C-ring propionate chain detaches from its conserved anchoring residues Ser272 and Ser274 (IX and X, **_Figure 3a_**). The strictly conserved His260 retracts from its position (XI and XII) and Tyr263 moves upward at 1 ps (XIII and XIV, **_Figure 3b_**). The water network connecting the C-ring propionate, the D-ring C = O, and His290 rearranges accordingly (**_Figure 3a_**). The excellent agreement between calculated and observed difference maps confirms these observations (**_Figure 3d and e_**, **_Figure 3—figure supplement 2_**). We conclude that the twist of the D-ring causes detachment of the C-ring propionate from the protein scaffold by dislocation of the C-ring, facilitated by the associated hydrogen bonding network.

Turning our attention to the B-ring, we find that the B-ring propionate breaks its salt bridge to Arg254 (**_Figure 3—figure supplement 3_**). However, this is not caused by movements of the chromophore backbone, as we observe little change on the B-ring itself. Instead, we find that a water bridge between the B- and C-ring propionates is broken as indicated by negative difference electron densities on the waters (**_Figure 3—figure supplement 3_**). Additionally, the highly conserved helix from Ser257 to Val269, moves away from the chromophore by an average of 0.36 Å in monomer A and 0.62 Å in monomer B (distances relative to the pyrrole water, **_Figure 3—figure supplement 4_**). The

**Table 1.** Crystallographic table.

| | Dark | one ps | 10 ps |
|---|---|---|---|
| PDB code | 6T3L | 6T3U | |
| Data collection | | | |
| Temperature (K) | 293 | 293 | 293 |
| Space Group | P212121 | P212121 | P212121 |
| Cell dimensions (a, b, c) | | | |
| a, b, c (Å) | 54.98 116.69 117.86 | 54.98 116.69 117.86 | 54.98 116.69 117.86 |
| $\alpha$, $\beta$, $\gamma$ (°) | 90.0 90.0 90.0 | 90.0 90.0 90.0 | 90.0 90.0 90.0 |
| Data resolution overall (Å)‡ | 45.77–2.07 | 41.46–2.21 | 45.77–2.14 |
| | (2.10–2.07) | (2.25–2.21) | (2.17–2.14) |
| $R_{split}$ (%)†‡ | 5.79 (120.05) | 10.59 (114.64) | 5.70 (121.86) |
| SNR (I/$\sigma(I)$)‡ | 9.21 (0.83) | 6.10 (0.88) | 10.11 (0.99) |
| CC(1/2)‡ | 0.99 (0.33) | 0.98 (0.38) | 0.99 (0.344) |
| Completeness (%)‡ | 100 (100) | 100 (100) | 100 (100) |
| Multiplicity‡ | 461.35 (65.9) | 106.11 (34.3) | 347.36 (62.1) |
| Number of hits | 149074 | 42853 | 159997 |
| Number of indexed patterns | 70726 | 21150 | 70335 |
| Indexing rate(%)@sectionsign | 47.44 | 49.35 | 43.96 |
| Number of total reflections | 24017763 | 5310179 | 17823530 |
| Number of unique reflections | 52060 | 39316 | 43279 |
| Refinement | | | |
| Resolution (Å)‡ | 45.82–2.07 | 36.94–2.21 | |
| | (2.12–2.07) | (2.27–2.21) | |
| $R_{work}$ / $R_{free}^{‡}$ | 0.162/0.191 | 0.230/0.256 | |
| | (0.317/0.346) | (0.411/0.443) | |
| Number of atoms | 5123 | 5135 | |
| Average B factor (Å2) | 76.44 | 78.63 | |
| R.m.s deviations | | | |
| Bond lengths (Å) | 0.007 | 0.006 | |
| Bond angles (°) | 1.251 | 1.152 | |

† $R_{split} = 1/\sqrt{2} \frac{\Sigma hkl |Ieven - Iodd|}{1/2 \Sigma hkl |Ieven + Iodd|}$ .

‡ Highest resolution shell is shown in parentheses.

§ Ratio of the number of indexed images to the total number of hits.

changes of the D-ring are transduced to Ser257 via the side chains of His260 and Tyr263, and as a result, the hydrogen bond of Ser257 to the B-ring propionate group breaks. The amino acids in the stretch are over 50% conserved (*Figure 3—figure supplement 4*), suggesting that it has evolved to transfer an ultrafast signal. We conclude that relaxation of the protein is necessary for the detachment of the B-ring propionate from the protein scaffold.

Next to the changes around the D-ring, the maps reveal strong difference electron density on the A-ring (XVIII and XIX), Asp207 (XX to XXIII) (*Figure 4a*) and the pyrrole water (XV) (*Figure 4b*). When interpreted and modelled as downward movement of the A-ring and Asp207 and photodissociation of the pyrrole water from the chromophore, excellent agreement between calculated and observed difference electron density is obtained (*Figure 4—figure supplement 1*). The A-ring is covalently attached to the protein backbone in phytochromes (*Song et al., 2014*), which renders complete isomerization impossible, but is sufficiently flexible to accommodate the proposed changes. The

**Table 2.** Difference electron density features listed for certain atoms.

| Object | Label | 1ps A | 1ps B | 10ps A | 10ps B |
|---|---|---|---|---|---|
| Pyrrole Water | | | | | |
| Pyrrole Water (-) | XV | −8.2σ | −9.4σ | −5.0σ | −6.5σ |
| Pyrrole Water (+) Alt. 1 | XVI | 4.8σ | 4.2σ | 3.0σ | 3.4σ |
| Pyrrole Water (+) Alt. 2 | XVII | 4.4σ | 5.8σ | 3.0σ | 3.3σ |
| D-ring | | | | | |
| D-ring N-H/C = O (-) | I | −4.2σ | −7.3σ | −6.4σ | −5.1σ |
| D-ring Methyl (-) | II | −4.3σ | −3.5σ | −2.9σ | −4.9σ |
| D-ring Vinyl (-) | III | −4.0σ | −4.8σ | −3.5σ | −3.4σ |
| D-ring N-H/C = O (+) | IV | 4.6σ | 6.4σ | 4.6σ | 4.1σ |
| D-ring Methyl (+) | V | 3.5σ | 3.4σ | - | - |
| D-ring Vinyl (+) | VI | 4.1σ | 4.6σ | 3.4σ | 2.6σ |
| C-ring | | | | | |
| C-propionate (-) | IX | −6.4σ | −5.8σ | −5.9σ | −4.7σ |
| C-ring (-) | VII | −5.5σ | −4.4σ | −5.1σ | −3.2σ |
| C-propionate (+) | X | 6.9σ | 6.4σ | 7.4σ | 4.9σ |
| C-ring (+) | VIII | 4.9σ | 5.2σ | 3.2σ | 5.4σ |
| A-ring | | | | | |
| A-ring C = O (-) | XVIII | −4.3σ | −4.7σ | −3.6σ | −2.9σ |
| A-ring C = O (+) | XIX | 5.2σ | 5.0σ | 5.7σ | 4.7σ |
| His260 | | | | | |
| His260 Sidechain(-) | XI | −6.7σ | −5.2σ | −6.3σ | −4.1σ |
| His260 Sidechain(+) | XII | 4.3σ | 4.5σ | 3.5σ | 3.2σ |
| Tyr263 | | | | | |
| Tyr263 Sidechain(-) | XIII | −6.9σ | −7.0σ | −4.7σ | −6.2σ |
| Tyr263 Sidechain(+) | XIV | 4.8σ | 4.9σ | 3.0σ | 4.3σ |
| Asp207 | | | | | |
| Asp207 Sidechain (-) | XX | −6.2σ | −6.7σ | −5.6σ | −3.4σ |
| Asp207 Sidechain(+) | XXI | 5.7σ | 5.0σ | 4.2σ | - |
| Asp207 Backbone (-) | XXII | −6.2σ | −5.4v | −5.1σ | −4.5σ |
| Asp207 Backbone (+) | XIII | 4.6σ | 4.4σ | 4.5σ | 3.5σ |
| Tyr176 | | | | | |
| Tyr176 Sidechain(-) | | −4.9σ | −3.9σ | −5.1σ | −3.8σ |
| Phe203 | | | | | |
| Phe203 Sidechain(-) | | −4.1σ | −4.6σ | −4.6σ | - |

pyrrole water may either move to feature (XVI), or occupy an anisotropic, worm-shaped feature which extends from the A-ring to the D-ring (XVII) (*Figure 4b*).

Furthermore, correlated negative and positive electron density features are observed on backbone atoms of the highly conserved stretch from Pro201 to His209. These difference electron density features indicate that the residues move away from the centre of the chromophore by an average of 0.54 Å and 0.57 Å in monomers A and B, respectively (*Figure 3—figure supplement 4b*). The stretch includes Asp207 and it is located between the A-ring and the PHY domain, which makes it plausible that the changes in the chromophore cause this protein rearrangement. The

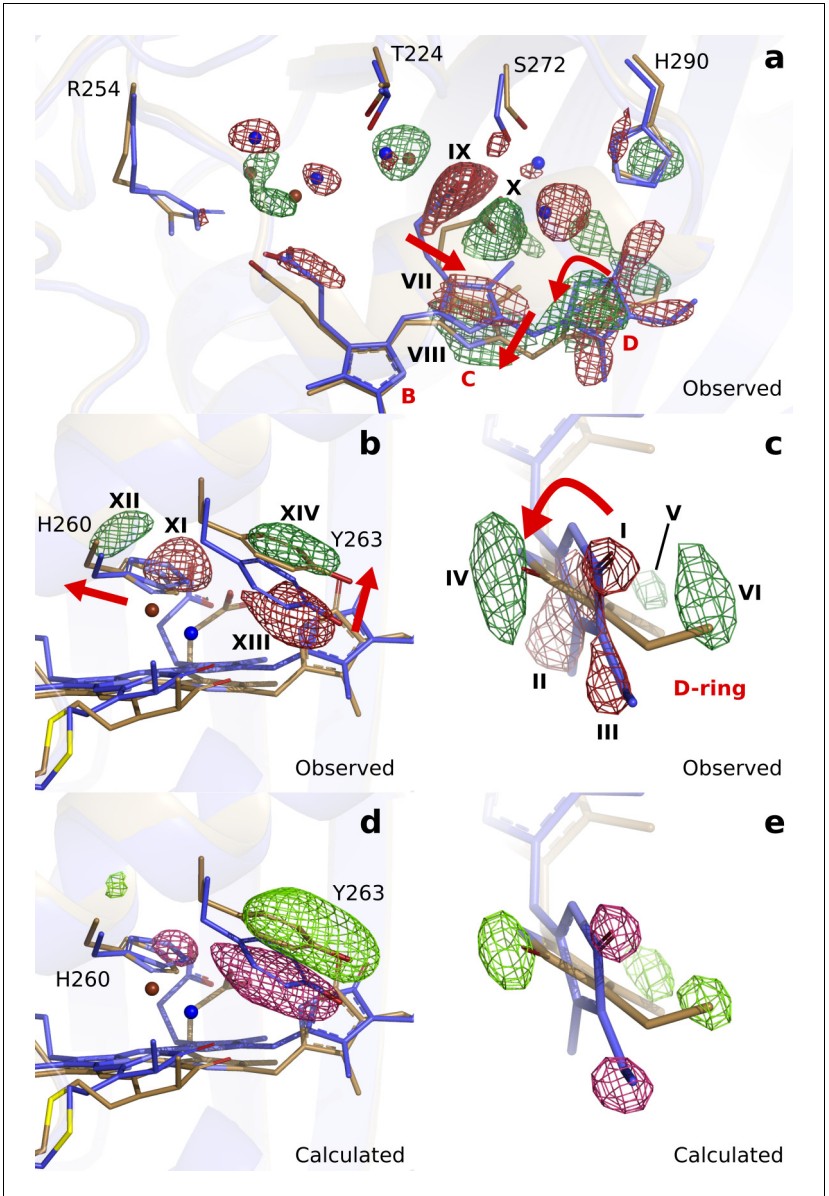

**Figure 3.** Observed and calculated difference electron densities reveal a twist of the D-ring and significant protein rearrangements. The observed difference electron density with the refined $Dr$BphP$_{dark}$(blue) and $Dr$BphP$_{1ps}$(beige) structures, shown for (**a**) the B-, C-, and D-ring surroundings, (**b**) the strictly conserved His260 and Tyr263, and (**c**) the D-ring. The calculated difference electron density shown for (**d**) His260 and Tyr263 and (**e**) the D-ring. The D-ring twists counter-clockwise when viewed along C15-C16 bond toward the C-ring. The observed difference electron density is contoured at 3.3 σ. And the calculated difference electron density is contoured at 3.5 and 5.0 σ for panel d and e, respectively. Monomer A is shown in this figure.

The online version of this article includes the following figure supplement(s) for figure 3:

**Figure supplement 1.** Extrapolated maps at $\alpha = 25$ demonstrate that the D-ring twists in the photoactivated state.

**Figure supplement 2.** Comparison of observed (upper panel) and calculated (lower panel) difference electron densities indicate good agreement around the B-D rings.

**Figure supplement 3.** Hydrogen bonding network between the chromophore propionate groups is disrupted at 1 ps delay time.

**Figure supplement 4.** Backbone movements at 1 ps.

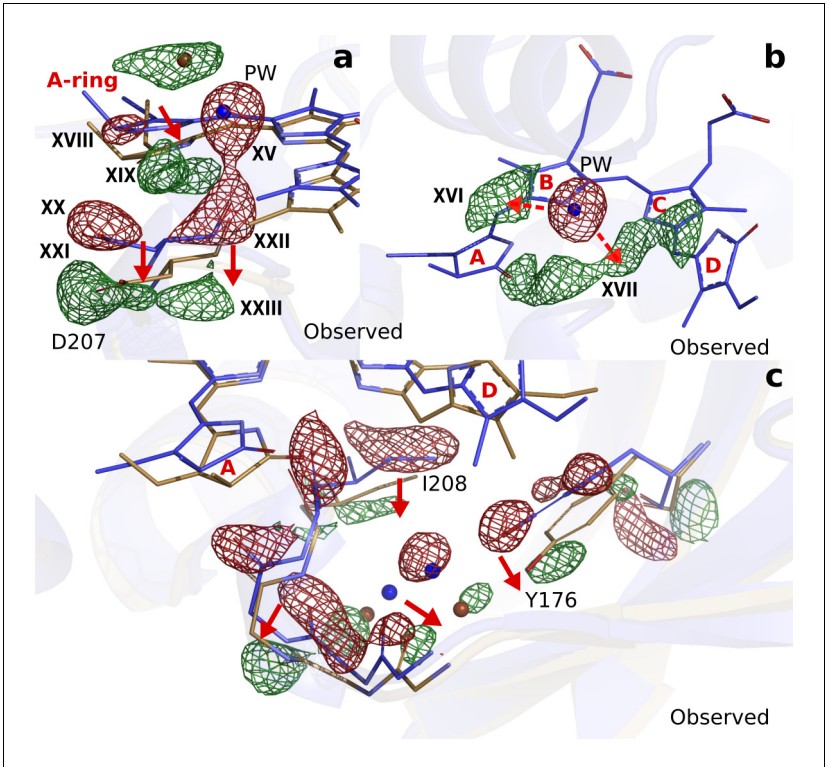

**Figure 4.** Photodissociation of the pyrrole water, displacement of the A-ring and its effect on the proteins scaffold. (a). The observed difference electron density displayed with the *Dr*BphP_dark (blue) and *Dr*BphP_1ps (beige) structures around the A-ring, Asp207 and pyrrole water (PW). The structural model was inconclusive as to whether the A-ring twists around the double bond between the B- and A-ring, or whether it tilts downward hinged on the connection between C- and B-ring. (b). The regions of the pyrrole water (PW) and the area between the pyrrole rings show negative and positive densities, respectively. The observed difference electron density is contoured at 3.3σ. (c). Density displayed for the backbone below the A-ring, including side chains of the strictly conserved Ile208 and Tyr176 as well as the surrounding water network. Monomer A is shown in this figure.

The online version of this article includes the following figure supplement(s) for figure 4:

**Figure supplement 1.** Comparison of observed and calculated difference electron densities in the A-ring region.

changes are complemented by significant rearrangements of a stretch of waters and a conserved Tyr176 (*Figure 4c*).

## Discussion

The structure of *Dr*BphP_CBD 1 ps after photoexcition reveals changes of the biliverdin chromophore and the surrounding residues. We find a twist of the D-ring, displacement of the C-ring, and associated changes of the water network which connects the D-ring, the C-ring propionate, and His290. Further, we identify a disruption of the salt bridge between the B-ring propionate and the Arg254, and significant changes around the A-ring, Asp207 and the pyrrole water. The changes are retained at 10 ps, even though at a lower population (*Figure 1—figure supplement 5*). The extensive and coordinated structural changes in the binding pocket (*Figure 3*) manifest a liberation of the chromophore from the protein scaffold, which we propose to be necessary for the conformational rearrangements to occur in the downstream photoconversion to Pfr.

Infrared spectral data indicate significant reorganization of the chromophore and several amino acids including the PHY-tongue region as early as in Lumi-R state, which is the first known ground state intermediate in the photoconversion from Pr to Pfr (*Ihalainen et al., 2018*; *van Thor et al., 2007*). However, the structure of the bilin and the binding pocket in Lumi-R is not known, because structural information is missing. Since the quantum yield of reaching the Lumi-R state is low (on the

order of 10%), spectroscopic investigation of the mechanism is difficult, and it is currently not fully established how the Lumi-R state is reached. Crystallographic data does not report on whether the chromophore is electronically excited or not, hence we cannot determine whether the structure that we observe is in a relaxed excited state or in a ground state. The time delay (1 ps) supports that our structure presents an intermediate enroute to the Lumi-R state.

The D-ring of the bilin chromophore isomerizes around the C15-C16 bond from (Z) in Pr to (E) in Pfr (*Rüdiger et al., 1983*; *Burgie et al., 2016*; *Takala et al., 2014*; *Yang et al., 2008*; *Essen et al., 2008*). Circular dichroism spectroscopy and solid-state NMR spectroscopy have indicated that the position of the D-ring inverts from an '$\alpha$'-facial (Pr) to a '$\beta$'-facial (Pfr) position in cyanobacterial and plant phytochromes, whereas it stays '$\alpha$'-facial in bacterial phytochromes (*Rockwell et al., 2009*; *Song et al., 2011*; *Song et al., 2018*). Based on anticipated steric clashes with the C-ring methyl group, it has been proposed that the D-ring rotates counter-clockwise in plant and cyanobacterial phytochromes, but clockwise in bacterial phytochromes (*Rockwell et al., 2009*). Moreover, spectroscopy has shown that the D-ring of the bilin chromophore is already isomerized in the Lumi-R state (*van Thor et al., 2007*; *Heyne et al., 2002*; *Yang et al., 2012*).

Seemingly contradictory, we now observe that the D-ring is rotated counter-clockwise by tens of degrees for the bacterial *Dr*BphP at 1 ps time delay (*Figure 3c and e*). The conformation is strongly supported by the difference map. It contains two positive peaks (V and VI in *Figure 3c*), which indicate the new position of the vinyl and methyl group of the D-ring. We tested models in which the D-ring was rotated in a clockwise direction, but the agreement with the experimental difference map decreased. Thus, it may be that the D-ring indeed rotates counter-clockwise in bacterial phytochromes, similar to plant and cyanobacterial phytochromes. For complete isomerization, this would mean that the C-ring moves out of the way during the rotation. We observe significant movements of the C-ring, which may be an indication for that such a mechanism is possible. Raising a note of caution, we cannot fully exclude that the truncation of our phytochrome construct or the crystal packing influences the direction of rotation. NMR studies have reported conformational heterogeneity in the chromophore binding pocket of phytochromes in solution (*Song et al., 2011*; *Lim et al., 2018*; *Song et al., 2018*; *Gustavsson et al., 2020*). Crystallization could select one of the conformations, which may have a preferred rotation in the counter-clockwise direction. More experiments are needed to clarify this question.

It is interesting to compare the structural changes at 1 ps time delay to the changes observed in the conversion between Pr and Pfr (*Burgie et al., 2016*; *Takala et al., 2014*; *Stojković et al., 2014*). Major changes include a flipped D-ring, changes in conserved residues of the chromophore-binding pocket, for example Tyr176, His201 and Phe203, and refolding of the PHY tongue. The PHY tongue is not included in our construct, but Tyr176 and Phe203 are associated with difference electron density features in our maps (*Table 2*). However, the movements are much smaller at 1 ps compared to the Pr-to-Pfr transition. This is not unexpected, given the short time delays, but it shows that the residues are tightly coupled to the chromophore. Interestingly, the Pr and Pfr structures also reveal a sliding movement of the entire chromophore (*Yang et al., 2011*; *Burgie et al., 2016*; *Takala et al., 2014*). This requires that the propionic groups have to break their bonds to the protein scaffold. Our data indicate that this is part of the primary photoresponse.

The photodissociation of the pyrrole water from the chromophore is a surprising finding. The pyrrole water is ubiquitously found in phytochrome structures (*Essen et al., 2008*; *Yang et al., 2008*; *Otero et al., 2016*; *Burgie et al., 2016*; *Wagner et al., 2005*; *Burgie et al., 2014*; *Schmidt et al., 2018*; *Yang et al., 2011*). Our fluence dependent SFX data show that the negative density on the pyrrole water is the last signal to disappear when lowering the photon excitation densities 10-fold (*Figure 2*). This makes us confident that the photodissociation reaction is not caused by multi-photon effects. The removal of the water requires significant energy, because the hydrogen bonds to the A-, B-, and C-rings of the chromophore and the backbone C = O group of Asp207 have to be broken. We do not think that the twist of the D-ring causes this through direct steric interactions, because there is no contact between the pyrrole water and the D-ring. Rather, it may be triggered by an excited state charge redistribution between the pyrrole water and the chromophore, for example by ultrafast proton or electron transfer (*Toh et al., 2010*). Such charge re-distributions are typically facilitated by changes in geometry (*Nosenko et al., 2008*) and may therefore be caused indirectly by structural changes of the A-, C-, or D-rings, but this requires further investigation.

Conformational changes of the A-ring, Asp207 and the pyrrole water have not been considered to occur on picosecond time scales. The strictly conserved Asp207 is a key residue for signal transduction because it connects the chromophore to the PHY-tongue in Pr and Pfr (*Essen et al., 2008*; *Yang et al., 2008*; *Takala et al., 2014*). Its displacement suggests, together with the relocation of the residue stretch surrounding it, that disruption of the GAF-PHY interface may occur as early as 1 ps after photoexcitation (*Figure 3—figure supplement 4b*). With a hydrogen bond to the pyrrole water and in tight steric contact with the A-ring, Asp207 thereby acts as an extended arm of the chromophore. We propose that the photodissociation of the pyrrole water from the bilin and the change of the A-ring are integral parts of ultrafast phytochrome signaling toward the PHY domain.

We demonstrate that within 1 ps, the D-ring twists, that the chromophore is liberated from the protein (*Figure 5a*) and that movements of the pyrrole water, the A-ring and Asp207 lead to signaling directed toward the PHY-tongue (*Figure 5b*). When mapped on the structure of the complete photosensory core module (*Takala et al., 2014*), both changes work together to destabilize the Arg466:Asp207 salt bridge. Tyr263 moves up, caused by the twist of the D-ring, and Asp207 moves down, caused by changes of the A-ring, retracting both residues from the salt bridge.

Our data reveal a highly collective primary photoresponse for phytochromes. This is consistent with the fact that most point mutations of conserved residues alter, but do not inhibit, photoconversion (*Wagner et al., 2008*). The ultrafast structural changes are more extensive than in bacteriorhodopsin, photoactive yellow proteins, and in a fluorescent protein (*Pande et al., 2016*; *Nogly et al., 2018*; *Coquelle et al., 2018*). While previously observed ultrafast backbone movements have been interpreted as 'protein quakes' for myoglobin and bacteriorhodopsin (*Barends et al., 2015*; *Nogly et al., 2018*; *Toh et al., 2010*), the present backbone motion in the phytochrome binding pocket are much more directed (*Figure 3—figure supplement 4*). The changes occur in highly conserved regions of the protein and are part of the collective signaling response of the entire binding pocket.

Phytochromes have to be able to stabilize the bilin and to direct its photoisomerization from two photochemical ground states, Pr and Pfr. These differ both structurally and electronically, which precludes a single reaction trajectory for isomerization in the two directions. With this in mind, the observed primary photoresponse is reasonable. The structural signal is highly delocalized already at 1 ps, causing near-simultaneous liberation of the chromophore and initial signal transduction. We propose that these reaction trajectories stabilize each other, navigating the protein into a productive

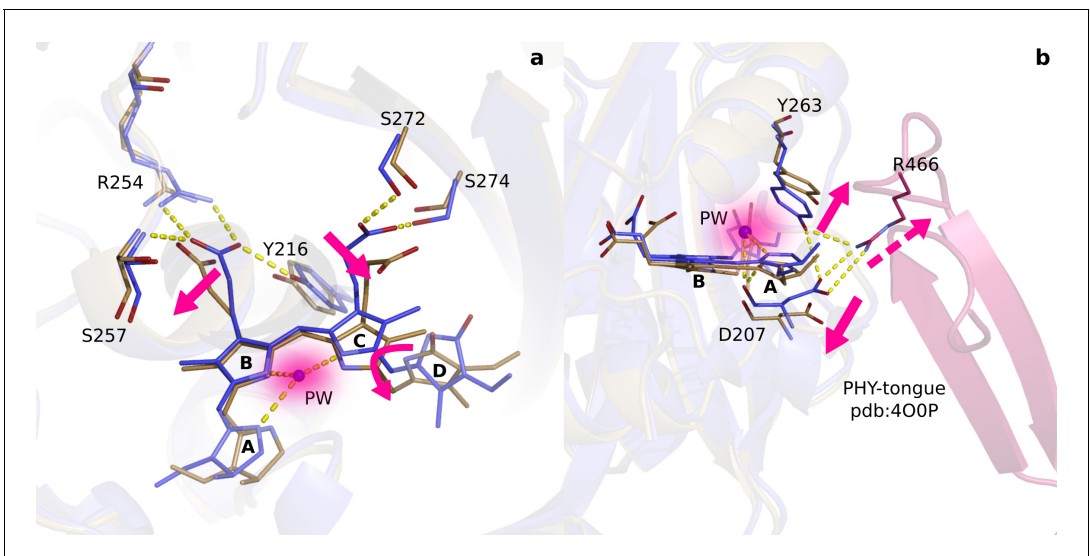

**Figure 5.** Two correlated photochemical events guide the primary photorespone of phytochrome proteins. (a). The structures (*Dr*BphP_dark, blue and *Dr*BphP_1ps, beige) indicate that rotation of the D-ring initiates breakage of non-covalent bonds of the propionates to the protein scaffold. Even the C- and A-rings are displaced significantly and the pyrrole water is dislocated from its original location at 1 ps (shade). (b). The same structures are overlayed with the complete photosensory core in Pr state (PDB ID 4O0P, pink) (*Takala et al., 2014*). The scissor-like separation of Asp207 and Tyr263 could result in breakage of the hydrogen bonds to Arg466 of the conserved PRXSF motif located in the PHY-tongue region.

reaction path. The multidimensional reaction trajectory is consistent with the low quantum yields for photoconversion (*Lamparter et al., 1997*), which are characteristic for the phytochrome superfamily. Whereas the twisting motion of the D-ring has been the working model for phytochrome activation and is now confirmed, the photodissociation of the pyrrole water is highly surprising. We propose that both chemical events work together and enable phytochrome proteins to translate light information into structural signals, guiding the growth and development of plants, fungi, and bacteria on Earth.

## Materials and methods

### Protein purification and crystallization

The $His_6$-tagged PAS-GAF domain from *D. radiodurans* (aa 1–321) in vector pET21b(+) (*Wagner et al., 2005*) was expressed and purified as previously described (*Lehtivuori et al., 2013*; *Takala et al., 2014*). The recombinant protein was expressed in *Escherichia coli* strain BL21(DE3), either with or without *Ho*1 to yield holo- or apoprotein, respectively. Cells were lysed with Emulsiflex and cleared by centrifugation (20,000 rpm, 30 min, +4˚C). Full biliverdin incorporation was ensured by adding 8 mg of biliverdin hydrochloride (Frontier Scientific) per litre of cell culture to the cell lysate, followed by overnight incubation on ice. The protein was then purified at room temperature with HisTrap HP column (GE Healtcare) in 30 mM Tris, 50 mM NaCl and 5 mM imidazole (pH 8) and eluted with increasing imidazole concentration (gradient elution over 5–500 mM). Size-exclusion chromatography was then conducted with a HiLoad 26/600 Superdex 200 pg column (GE Healthcare) in buffer (30 mM Tris pH 8.0). Finally, the protein was concentrated to 30–50 mg/mL and flash-frozen in liquid nitrogen.

Crystals were set up under green safe light and grown in dark. Batch crystallization was performed as described (*Edlund et al., 2016*). 50 µL of purified protein (25–30 mg/mL) was added to 450 µL of reservoir solution (60 mM Sodium acetate pH 4.95, 3.3% PEG 400, 1 mM DTT and 30% 2-methyl-2,4-pentanediol) and immediately mixed. Initial microcrystals were grown on a tipping table at 4 ˚C for 48 hr. Once the microcrystals were formed, additional protein was added to increase crystal size. The microcrystals were first pelleted by brief centrifugation and 400 µL of supernatant was removed. 200 µL of diluted protein (14 mg/mL in 30 mM Tris pH 8.0) was then added to the microcrystals along with 200 µL of fresh reservoir solution. After 48 hr incubation on a tipping table at room temperature, crystals of diffraction quality (20–70 µm long needles) were formed (*Figure 1—figure supplement 2* and *Table 1*).

### Transient absorption experiment of microcrystals

Transient absorption experiments were performed on a home build setup based on a Ti:sapphire femtosecond laser system (1 kHz, 800 nm). The main beam was split into pump and probe beams. The pump beam was sent through the home build noncollinear optical parametric amplifier to produce excitation pulses at 640 nm central wavelength. The probe beam was focused on a 2 mm sapphire plate to generate broadband (400–760 nm) white light which was split by 50/50 beamsplitter to reference and probe beams. The mutual polarization of the pump and probe beams was set to the magic angle (54.7˚) by Berek compensator. The probe beam was focused on a sample cuvette that was continuously translated in vertical axis to prevent sample degradation. The microcrystals were washed with crystallization buffer five time in order to remove the solubilized proteins. 2.5 µL of microcrystals including a small amount of crystallization buffer were placed between two $CaF_2$ windows without a spacer. The OD of the sample was about 0.6. Time-resolved absorption changes were measured by detecting probe and reference beams dispersed on the double-diode array; the time delay between pump and probe pulses was set by a computer controlled delay line placed in the probe beam path. All measurements were carried out in room temperature.

### Light scattering of the grease jet

In order to estimate the light intensity of the optical laser in the grease jet, the optical transmission of the grease, grease mixed with crystallization buffer, microcrystals in the grease matrix and pure microcrystals were measured with a transmittance diode-array UV-Vis spectrometer (Cary 8454, Agilent Technologies) (*Figure 1—figure supplement 1c*). 2.5 µl each of sample was placed between

two CaF$_2$ windows with a 50 µm Teflon spacer and squeezed together, resembling the characteristics of the jet during the XFEL experiments. The raw spectra of microcrystals (*Figure 1—figure supplement 1c*) in the buffer has been measured between two CaF$_2$ windows without spacer to minimize the absorption loss, the pathlength was estimated to be $\leq 50$µm.

## Fluence calculations

The optical laser parameters used for the experiment were as follows: wavelength was 640 nm, the laser-spot dimensions at the focus was $100 \times 80$ µm² FWHM ($170 \times 136$ µm² at $1/e^2$ intensity), the pulse energy was 40 µJ, the nominal pulse duration was 70 fs (not confirmed at the sample position), and the repetition rate was 30 Hz. The energy of a photon at 640 nm is $3.1 \times 10^{-19}$ J. Using the photon density of the laser at $1/e^2$ convention of 1.7 mJ/mm², we obtain a photon fluence of $5.48 \times 10^{15}\, \mathrm{photons \cdot mm^{-2}}$. The extinction coefficient of biliverdin in the phytochrome at 640 nm ($\epsilon_{640}$) is $27.7 \times 10^3 \sim \mathrm{M^{-1} cm^{-1}}$ and the cross section is then $\sigma_{640} = ln(10) \cdot \epsilon_{640} \cdot 1000/\mathrm{N_A} = 1.06 \times 10^{-14} \sim \mathrm{mm^2 \cdot molecule^{-1}}$, where $N_A$ is Avogadro's number. Multiplying the photon fluence with the cross section yields 58 photons per molecule. Light scattering in the carrier matrix decreases the effective fluence of photons that interact with the crystals. Our absorption spectra of the grease (*Figure 1—figure supplement 1*) indicate that the grease is transparent when untreated, but attenuates the light intensity by 2 orders of magnitude in the visible region when mixed with crystallization buffer or crystals (pathlength 50 µm). This indicates that almost every photon is scattered. Therefore, even when neglecting the scattering of the jet surface, crystals will be exposed to a photon fluence that is significantly reduced. A reduction of the photon fluence by 2 orders of magnitude is a realistic assumption as we used grease jets with a diameter of 75 µm or 100 µm. Another factor that contributes to the reduction of the number of photons per chromophore and non-homogeneous illumination of the microcrystals is the orientation of the crystals and the high chromophore density in them. The first few chromophores in the light path will shade the remaining chromophores in the needle-shaped crystals. Since the X-rays probe every molecule in their path with approximately same likelihood, the average photon fluence per probed chromophore is reduced. Assuming that the effective fluence inside the grease jet is reduced by a factor of 100, we estimate an average number of 0.5–1 photons per chromophore. This is consistent with the photoexcitation yield of 8% and with our experimental finding that the difference signal vanishes under the noise signal when reducing the photon fluence by a factor of 10 (*Figure 2*).

## SFX data acquisition

Serial femtosecond crystallographic data were collected at SPring-8 Angstrom Compact Free Electron Laser (SACLA) in two beamtimes in October 2018 and May 2019. The microcrystals were pelleted by brief centrifugation and the crystal pellet was mixed with 180 µL of grease. The grease/crystal mixture was loaded into a 4 mm sample reservoir for data acquisition. The sample was delivered to the X-ray beam at a flow rate of 2.5 µL/min or 4.2 µL/min for 75 µm and 100 µm diameter nozzles, respectively. The time resolution of the experiment was limited by the jitter of the XFEL of 100 fs *r.m.s.* The experimental settings were nominally the same for the 1 ps and 10 ps delay times and all data were recorded during 7 hr of beamtime. We also recorded data at 3 ps delay time, but these generated electron density maps of poor quality due to an unknown reason and were therefore not analyzed further.

## Data processing

The background of the detector was estimated by averaging the first 150 dark images in each run and then subtracted from each diffraction pattern. Diffraction images with Bragg spots (the 'hits') were found by a version of Cheetah adapted for SACLA (*Nakane et al., 2016*; *Barty et al., 2014*). These hits were indexed by the program CrystFEL (version 0.6.3) (*White et al., 2012*). Indexing was performed using Dirax and Mosflm (*Duisenberg, 1992*; *Battye et al., 2011*). Spot finding in each diffraction image was done with the peakfinder8 algorithm using the parameters (min SNR = 4.5, threshold = 100, minimum pixel counts = 3). The indexed patterns were merged and scaled using partialator in CrystFEL and hkl files were produced. The figure of merits (*Table 1*) were calculated by using compare_hkl and check_hkl in CrystFEL. The histograms of the unit cell parameters are presented in *Figure 3*. All diffraction images have been deposited to CXIDB under ID 121.

### Refinement of dark structure

The initial phases were solved by molecular replacement with Phaser (*McCoy et al., 2007*) and the PAS-GAF crystal structure (PDB ID 5K5B) (*Edlund et al., 2016*) as a search model. The structure was refined with REFMAC version 5.8.0135 (*Murshudov et al., 2011*) with a weight factor for the geometry restraints of 0.05, accompanied by model building steps with Coot 0.8.2 (*Emsley et al., 2010*). The final structure (*Dr*BphP$_{dark}$) had Rwork/Rfree of 0.161/0.192 and no Ramachandran outliers (*Table 1*). The coordinates and structure factors have been deposited in the Protein Data Bank under the accession code 6T3L.

### Computation of difference electron density maps

The difference structure factors ($\Delta F$) are computed from the measured structure factor amplitudes in dark and for preset delay times between laser and X-ray pulses as $|\Delta F_o| = w(|F_o^{light}| - |F_o^{dark}|)$ and with phases taken from the dark structural model (*Dr*BphP$_{dark}$). $|F_o^{dark}|$ and $|F_o^{light}|$ were brought to the absolute scale by first scaling $|F_o^{dark}|$ to $|F_c^{dark}|$ and then scaling $|F_o^{light}|$ to $|F_o^{dark}|$ using the CCP4 program Scaleit (*Winn et al., 2011*). Difference Fourier density maps were calculated with a low resolution scaling cut-off at 18 Å . A weighting factor ($w$) was determined for each reflection to reduce the influence of outliers (*Ren et al., 2001*). From the weighted $\Delta F$, a difference electron density map ($\Delta\rho$) is calculated using the program 'fft' from the ccp4 suite of programs (*Winn et al., 2011*). Since $\Delta F$ are on the absolute scale, $\Delta\rho$ is on half the absolute scale as a result of the difference Fourier approximation (*Henderson and Moffat, 1971*; *Pandey et al., 2020*).

### Structure refinement of light structure

Extrapolated structural factors were assembled from amplitudes computed as $|F_e| = |F_c^{dark}| + \alpha * |\Delta F_o|$. The $F_c^{dark}$ denotes the calculated structure factors of the refined dark structure (*Dr*BphP$_{dark}$). The phases were taken from *Dr*BphP$_{dark}$ is inversely related to the population of the photoinduced state by $(100/\alpha) * 2$ (*Pandey et al., 2020*; *Henderson and Moffat, 1971*). We estimated $\alpha$ based on $F_e$ map features in the chromophore-binding pocket. Too high values for $\alpha$ lead to physically unrealistic negative electron density. We converged to $\alpha = 25$, which corresponds to 8% photoexcitation yield.

$F_e$ represents the pure structure factor of the photo-activated state (*Figure 3—figure supplement 1*). Refinement of a structural model was then performed in real and reciprocal space, using Coot (*Emsley et al., 2010*) and Phenix (*Adams et al., 2010*). The equilibrium values for the restraints used in the refinement of the biliverdin chromophore were taken from a minimal energy biliverdin ground state geometry that was obtained at the B3LYP/6–31G* level of density functional theory. Torsional restraints for the excited state geometry with the twisted D-ring were obtained at the SA (5)- CASSCF(12,12)/cc-pVDZ level of ab initio theory. We removed the torsion restrains for the C/D-ring (C14-C15-C16-C17; C14-C15-C16-ND; C13-C14-C15-C16; NC-C14-C15-C16) and for the A/B-ring (C3-C4-C5-C6; NA-C4-C5-C6; C4-C5-C6-C7; C4-C5-C6-NB) during refinement. The overall aim of the refinement was to maximize the agreement between the observed and calculated difference maps. To evaluate the agreement, we subtracted the calculated from the observed difference electron density ($\Delta\rho_o - \Delta\rho_c$). The computation of this difference-difference maps require scaling of the maps to each other. To do so, the highest and lowest intensities of $\Delta\rho_o$ were scaled to the corresponding maximum and minimum of $\Delta\rho_c$ and the observed $\Delta\rho_o$ were interpolated linearly according to this scaling. The resulting difference-difference electron density map was used to identify sites, which required further optimization in subsequent refinement steps. Calculation of Pearson Correlation Coefficient (PCC) values between the $\Delta\rho_o$ and $\Delta\rho_c$ were applied to guide refinement of specific regions, such as the D-ring and the whole chromophore region. To do so, the correlation was determined based on electron density within a sphere with a radius of 3.5 Å or 10 Å centred on the D-ring or pyrrole water, respectively. As a final step in the refinement procedure, we refined the models with REFMAC version 5.8.0135 (*Murshudov et al., 2011*) with high geometry restraints (weight factor 0.005). This was done against phased extrapolated structure factors, using the phases of the refined light and dark structure for computation of phased $\Delta F$ as described (*Pande et al., 2016*). The structures did not change much, although the R factors dropped in this last step of refinement to Rwork/Rfree of 0.230/0.256 (*Table 1*). The coordinates and structure factors have been deposited in the Protein Data Bank under the accession code 6T3U.

## Acknowledgements

The experiments at SACLA were performed at BL3 with the approval of the Japan Synchrotron Radiation Research Institute (JASRI) (Proposal No. 2018A8055 and 2019A8007). SW acknowledges the European Research Council for support (grant number: 279944). This work was supported by Academy of Finland grants 285461 and 296135 (HT and JAI, respectively) and Jane and Aatos Erkko foundation (JAI). This research is partially supported by Platform Project for Supporting Drug Discovery and Life Science Research (Basis for Supporting Innovative Drug Discovery and Life Science Research (BINDS)) from Japan Agency for Medical Research and Development (AMED). We thank Dr. Takanori Nakane for assistance with data processing during the beamtime and Heli Lehtivuori for the spectroscopic measurements with the microcrystals. This work was supported by NSF Science and Technology Centers grant NSF-1231306 ('Biology with X-ray Lasers') (MS), the National Science Foundation (NSF)-MCB-RUI 1413360 and NSF-MCB-EAGER 1839513 Research Grants to EAS. This work has been done as part of the BioExcel CoE (www.bioexcel.eu), a project funded by the European Union contracts H2020-INFRAEDI-02-2018-823830 and H2020-EINFRA-2015-1-675728 (GG).

## Additional information

### Funding

| Funder | Grant reference number | Author |
|---|---|---|
| European Research Council | 279944 | Sebastian Westenhoff |
| Academy of Finland | 285461 | Heikki Takala |
| Academy of Finland | 296135 | Janne A Ihalainen |
| Jane and Aatos Erkko Foundation | | Janne A Ihalainen |
| National Science Foundation | (NSF)-MCB-RUI 141336 | Emina A Stojković |
| National Science Foundation | NSF-MCB-EAGER 183951 | Emina A Stojković |
| Horizon 2020 | H2020-INFRAEDI-02-2018-82383 | Gerrit Groenhof |
| Horizon 2020 | H2020-EINFRA-2015-1-67572 | Gerrit Groenhof |
| National Science Foundation | NSF-1231306 | Marius Schmidt |

The funders had no role in study design, data collection and interpretation, or the decision to submit the work for publication.

### Author contributions

Elin Claesson, Planned experiments, Purified the protein and prepared the microcrystals, Acquired the data at SACLA, Computed difference electron density maps and refined the structural models, Wrote the paper with input from all authors; Weixiao Yuan Wahlgren, Heikki Takala, Purified the protein and prepared the microcrystals, Acquired the data at SACLA, Computed difference electron density maps and refined the structural models; Suraj Pandey, Acquired the data at SACLA and processed the crystallography data; Leticia Castillon, Purified the protein and prepared the microcrystals, Acquired the data at SACLA; Valentyna Kuznetsova, Acquired the data at SACLA, Measured the UV-Vis spectroscopic data; Léocadie Henry, Matthijs Panman, Melissa Carrillo, Joachim Kübel, Rahul Nanekar, Linnéa Isaksson, Amke Nimmrich, Andrea Cellini, Michał Maj, Robert Bosman, Eriko Nango, Rie Tanaka, Tomoyuki Tanaka, Luo Fangjia, So Iwata, Shigeki Owada, Emina A Stojković, Acquired the data at SACLA; Dmitry Morozov, Gerrit Groenhof, Provided the geometry restrains biliverdin in the excited state; Moona Kurttila, Measured the UV-Vis spectroscopic data; Keith Moffat, Conceived the experiments; Janne A Ihalainen, Conceived the experiments, Acquired the data at SACLA; Marius Schmidt, Conceived the experiments, Acquired the data at SACLA, Computed difference electron density maps and refined the structural models, Wrote the paper with input from all authors; Sebastian Westenhoff, Conceived the experiments, Planned experiments, Acquired the

data at SACLA, Computed difference electron density maps and refined the structural models, Wrote the paper with input from all authors

## Author ORCIDs

Weixiao Yuan Wahlgren (iD) https://orcid.org/0000-0003-0413-165X
Heikki Takala (iD) https://orcid.org/0000-0003-2518-8583
Matthijs Panman (iD) http://orcid.org/0000-0003-3853-123X
Janne A Ihalainen (iD) https://orcid.org/0000-0002-8741-1587
Sebastian Westenhoff (iD) https://orcid.org/0000-0002-6961-8015

## Decision letter and Author response

Decision letter https://doi.org/10.7554/eLife.53514.sa1
Author response https://doi.org/10.7554/eLife.53514.sa2

# Additional files

## Supplementary files

• Transparent reporting form

## Data availability

Crystallography data have been submitted to protein data bank (PDB) dark:ID: D_1292104678 and PDB ID: 6T3L 1ps:ID: D_1292104679 and PDB ID: 6T3U. Raw diffraction images are deposited under ID 121 at CXIDB.

The following datasets were generated:

| Author(s) | Year | Dataset title | Dataset URL | Database and Identifier |
|---|---|---|---|---|
| Claesson E, Wahlgren WY, Takala H, Pandey S, Castillon L, Kuznetsova V, Henry L, Panman M, Carrillo M, Kübel J, Nanekar R, Isaksson L, Nimmrich A, Cellini A, Morozov D, Maj M, Kurttila M, Bosman R, Nango E, Tanaka R, Tanaka T, Fangjia L, Iwata S, Owada S, Moffat K, Groenhof G, Stojković EA, Ihalainen JA, Schmidt M, Westenhoff S | 2020 | The primary structural photoresponse of phytochrome proteins captured by a femtosecond X-ray laser | http://cxidb.org/id-121.html | CXIDB, 10.11577/1607859 |
| Claesson E, Takala H, Wahlgren W, Pandey S, Schmidt M, Westenhoff S | 2020 | PAS-GAF fragment from Deinococcus radiodurans phytochrome in dark state | http://www.rcsb.org/structure/6T3L | RCSB Protein Data Bank, 6T3L |
| Claesson E, Takala H, Wahlgren W, Pandey S, Schmidt M, Westenhoff S | 2020 | PAS-GAF fragment from Deinococcus radiodurans phytochrome 1ps after photoexcitation | http://www.rcsb.org/structure/6T3U | RCSB Protein Data Bank, 6T3U |

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
