## [Decision Letter]

**Acceptance summary:**

The paper reports an excellent application of free electron laser crystallography to a biologically important problem. Only few such applications, all on different systems, have been published so far. The results reveal the detailed mechanism at picosecond and atomic resolution by which rearrangement of the chromophore in a phytochrome enable plants, bacteria and fungi to sense red light.

**Decision letter after peer review:**

Thank you for submitting your article "The primary structural photoresponse of phytochrome proteins captured by a femtosecond X-ray laser" for consideration by *eLife*. Your article has been reviewed by two peer reviewers, and the evaluation has been overseen by a Reviewing Editor and Cynthia Wolberger as the Senior Editor. The reviewers have opted to remain anonymous.

The reviewers have discussed the reviews with one another and the Reviewing Editor has drafted this decision to help you prepare a revised submission.

Summary:

In their manuscript Claesson et al. describe time-resolved crystallographic measurements to visualize picosecond structural changes in the ligand-binding domain of a bacterial phytochrome. Such photosensors allow plants, fungi and bacteria to detect light and adapt their physiological responses. The actual photochemical reaction happens in the ultrafast time regime at an bilin cofactor which later transmits the energy into conformational changes of the protein. It is quite remarkable how much the protein has already changed within a picosecond after a photon has been absorbed.

The only way to structurally resolve such ultrafast reactions is using one of the new X-ray laser facilities. These provide intense X-ray pulses with a few tens of fs in lengths, which in combination with femtosecond pump laser allows to resolve high-resolution structural snapshots at precise time delays after photoactivation. The method is very powerful but demanding and by far not routine, with only very few published examples.

This paper describes the first use of an XFEL in a pump-probe experiment to monitor the photoactivation of a phytochrome, in this case the *Dr*BphP(PAS-GAF) bidomain. The authors show that the B- and C-ring propionates loose contact with the protein, that the pyrrole water disappears, the D-ring rotates anti-clockwise and that the A- and B-rings move downwards within 1 ps of photon absorption. Most of these observations are unexpected.

The phytochrome reaction described here is clearly different from previous XFEL studies and a very valuable addition to our knowledge on how organisms sense and use light. The primary photoresponse in a phytochrome is of considerable interest and the structural details described in the manuscript will find a wide readership.

The manuscript was a pleasure to read and the study is overall well done.

Essential revisions:

1) The authors provide PDB file coordinates etc. for the dark and 1 ps datasets but not for that collected at 10 ps. Why not? As the authors show the relevant difference maps, they must have a structural solution too. This additional information is important and would enhance the scientific value of the paper considerably.

2) The authors describe anticlockwise rotation of the D-ring at 1 ps. They should address the fact that this contradicts the time-resolved IR polarization data that indicated D-ring rotation at 30 ps and that a clockwise rotation had been expected.

3) The authors must comment on the differences between the 1 ps (and 10 ps!) structure/s and that of the final Pfr state (which is not reached in this PAS-GAF construct). For example, the D-ring rotation they show is only partial (in Pfr the D-ring rotates much further), there appear to be no changes in the aromatic side groups that attend ring D (there a major differences in Pfr) and, most importantly, not only are the A- and B-rings positioned similarly in Pr and Pfr, the pyrrole water in seen in both (implying that the differences described in this paper represent a push-button effect – which might of course be correct).

4) Rather sensibly, the authors say they are unsure if the 1 ps structure is still in S1 or if it is a ground state. However, in the supplementary information they say explicitly that they "demonstrate that the D-ring twists in the excited state". Whereas this is quite likely to be true (and has been suggested earlier), they don't demonstrate it.

5) The authors suggest a possible "ultrafast hydrogen or electron transfer" charge redistribution model to explain their observations. It would make more sense to say "proton" rather than "hydrogen".

6) There are additional issues surrounding the fluence rates used. Strong scattering might complicate things, but that is just a problem the authors have with their methods. The difference signals they report continue to increase between the two highest fluences used, implying that – even in microcrystals – there will be significant light gradients within each crystal. Is this a problem? On the other hand, the fluence rates are likely to be very high despite the scattering, whereby the classical problem of multi-photon events arises. This reviewer does not think that the statement that "lowering the excitation density resulted in a joint reduction of all difference signals" is entirely correct. Given that the authors plainly state that "The overall aim of the refinement was to maximise the agreement between the observed and calculated difference maps", they should discuss the distinct possibility of artefacts more thoroughly.

7) The laser fluence is a critical parameter not only for conducting but also for evaluating the results of ultrafast time-resolved crystallographic experiments. The authors are very aware of this and have attributed much of the supplementary information and some of the available beamtime to this issue. However, much of their discussion is based on calculations and assumption and the reviewer could not find the original values used during the experiments. Please add the used laser pulse energy, spot size and pulse duration to the Materials and methods section describing fluence calculations, so the interested reader can follow the argumentation better.

8) The missing parameters aside, the stated value of 0.65 photons per molecule is based on many assumptions including a correction from displacement of the laser (only mentioned in Figure 2) and a factor of 100 through scattering from the grease (Figure 1—figure supplement 1 states "at least a two orders of magnitude… depending on the mixing procedure"). In truth, we do not know how many photons per molecule reach the chromophore and this can only ever be an average in crystals anyway. The reviewer suggests that the authors replace the value of 0.65 photons per molecule by a range to better reflect this uncertainty. The authors should also think about replacing the statement "the excitation density was in the one photon regime" used in the main text and Materials and methods section with an expression that is less matter of fact. Importantly this reviewer does not think that this uncertainty does devaluate the biological interpretation of the data since critical features including photodissociation of the pyrrol water occur at much lower laser fluence. The higher photon densities just lead to better maps by increasing activation levels.

9) In a related issue, one reviewer finds it strange that the difference maps for the 10 ps data are much weaker compared to the ones at 1 ps despite better statistics in terms of multiplicity and to some extent also resolution based on the crystallographic Table 1. There is no reason given in the manuscript for the lower intensity of the difference map features. Is the initial photoactivated state just decaying that rapidly (possible but my guess is no) or were different pump-probe settings used for the 10 ps data? Please include any changes in the experimental settings between the two delays in the manuscript.

10) The 1 ps difference map is of high quality even though the authors state low quantum yields for the photoactivation of phytochromes in the order of 10%. Capturing low activation levels reliably is an important advantage of the technique and this study demonstrates this nicely. However, it would be interesting to have an estimate for the achieved levels of activation within the crystal to see how close they get to the stated quantum yields.

---

## [Author Response]

Essential revisions:1) The authors provide PDB file coordinates etc. for the dark and 1 ps datasets but not for that collected at 10 ps. Why not? As the authors show the relevant difference maps, they must have a structural solution too. This additional information is important and would enhance the scientific value of the paper considerably.

Unfortunately, we were not successful in generating a satisfactory structural model against the data at 10ps. This structural model would not provide significant scientific insight beyond what we state in the paper, i.e. that the structural changes are similar at 1ps and 10ps. However, we demonstrate this statement more directly with Figure 1—figure supplement 5, which compares the difference densities at 1ps and 10 ps in detail.

The reason for the low reliability of a structural model at 10 ps is the following: The difference maps for 1ps and 10ps show similar features, but the signal strength is lower at 10ps, indicating that the population is lower at 10ps compared to 1ps (see also comment 9). A general restraint in time-resolved crystallography is that the measured amplitudes of the structure factor after laser excitation contain a mixture of molecules in the resting and activated state. The *F_o_-F_o_* difference maps presented in the paper are the most unbiased presentation of the structural changes between the resting and excited state. Refining a structure requires the reconstruction of the structure factor of the "pure" activated state, i.e. by removing the contribution of the resting state molecules to the data. This can be done, but introduces uncertainty because the photoactivation yield has to be estimated. We are confident that we got this right for the 1ps structures (8% yield corresponding to α = 25, see below and Materials and methods section), but with the 10ps structure the uncertainty would be higher, since the population of the photoactivated state is even lower. Given an expected low reliability of the model, refining a structural model would not provide a significant scientific insight beyond the finding that the structural changes are similar at 1ps and 10ps.

We have written a more elaborate justification in the revised manuscript, in the first paragraph of the Discussion, and in the Materials and methods section. We have also strengthened the statement that the structural changes at 1ps are retained at 10ps in the main text and added Figure 1—figure supplement 5.

2) The authors describe anticlockwise rotation of the D-ring at 1 ps. They should address the fact that this contradicts the time-resolved IR polarization data that indicated D-ring rotation at 30 ps and that a clockwise rotation had been expected.

We thank the reviewers for pointing this out! In the revised version, we now discuss the direction of the rotation more extensively, in particular with respect to Karsten Heyne’s IR work and the CD spectra of Rockwell and Lagarias. The direction of motion may be influenced by the truncated PAS-GAF construct that we use (PHY domain missing) and by crystal packing. To the best of our knowledge, the direction of rotation has not been observed directly but it has been inferred from steric arguments and from CD/IR measurements that indicate the end states. Unfortunately, a structure of a plant phytochrome or cyanobacterial phytochrome in the light state is not available. This is now discussed in the Discussion.

3) The authors must comment on the differences between the 1 ps (and 10 ps!) structure/s and that of the final Pfr state (which is not reached in this PAS-GAF construct). For example, the D-ring rotation they show is only partial (in Pfr the D-ring rotates much further), there appear to be no changes in the aromatic side groups that attend ring D (there a major differences in Pfr) and, most importantly, not only are the A- and B-rings positioned similarly in Pr and Pfr, the pyrrole water in seen in both (implying that the differences described in this paper represent a push-button effect – which might of course be correct).

We thank the reviewers for this comment. We have inserted a paragraph in the Discussion; it provides interesting new thoughts to the paper.

4) Rather sensibly, the authors say they are unsure if the 1 ps structure is still in S1 or if it is a ground state. However, in the supplementary information they say explicitly that they "demonstrate that the D-ring twists in the excited state". Whereas this is quite likely to be true (and has been suggested earlier), they don't demonstrate it.

We thank the reviewers for pointing out this inconsistency, and have corrected the sentence in the supplementary information accordingly.

5) The authors suggest a possible "ultrafast hydrogen or electron transfer" charge redistribution model to explain their observations. It would make more sense to say "proton" rather than "hydrogen".

We agree and have changed the word to 'proton'.

6) There are additional issues surrounding the fluence rates used. Strong scattering might complicate things, but that is just a problem the authors have with their methods. The difference signals they report continue to increase between the two highest fluences used, implying that – even in microcrystals – there will be significant light gradients within each crystal. Is this a problem? On the other hand, the fluence rates are likely to be very high despite the scattering, whereby the classical problem of multi-photon events arises. This reviewer does not think that the statement that "lowering the excitation density resulted in a joint reduction of all difference signals" is entirely correct. Given that the authors plainly state that "The overall aim of the refinement was to maximise the agreement between the observed and calculated difference maps", they should discuss the distinct possibility of artefacts more thoroughly.

We thank the reviewers for the comments on photon fluence. We address them below in response to reviewers’ comment 8.

7) The laser fluence is a critical parameter not only for conducting but also for evaluating the results of ultrafast time-resolved crystallographic experiments. The authors are very aware of this and have attributed much of the supplementary information and some of the available beamtime to this issue. However, much of their discussion is based on calculations and assumption and the reviewer could not find the original values used during the experiments. Please add the used laser pulse energy, spot size and pulse duration to the Materials and methods section describing fluence calculations, so the interested reader can follow the argumentation better.

We fully agree and have added the original parameters in the Materials and methods.

8) The missing parameters aside, the stated value of 0.65 photons per molecule is based on many assumptions including a correction from displacement of the laser (only mentioned in Figure 2) and a factor of 100 through scattering from the grease (Figure 1—figure supplement 1 states "at least a two orders of magnitude… depending on the mixing procedure"). In truth, we do not know how many photons per molecule reach the chromophore and this can only ever be an average in crystals anyway. The reviewer suggests that the authors replace the value of 0.65 photons per molecule by a range to better reflect this uncertainty. The authors should also think about replacing the statement "the excitation density was in the one photon regime" used in the main text and Materials and methods section with an expression that is less matter of fact. Importantly this reviewer does not think that this uncertainty does devaluate the biological interpretation of the data since critical features including photodissociation of the pyrrol water occur at much lower laser fluence. The higher photon densities just lead to better maps by increasing activation levels.

Indeed, there is likely significant inhomogeneity in the photoexcitation of the crystals. We have therefore changed the number to a sensible range: 0.5 – 1 photons/molecule. We have also added a more detailed explanation of how we arrive at the number/range, stating the original parameters and the factors that we believe should be considered when evaluating photon fluence in grease jets.

We agree to the reviewers’ view that the statement about the one-photon regime was too categorical. We have therefore changed it to “Critical signals, like the twist of the D-ring and the photodissociation of the pyrrole water sustained when lowering the excitation densities, indicating that the signal arises predominantly from one-photon excitation.”.

A related change in the manuscript is that we now state in the legend of Figure 1—figure supplement 1 that we tried to reproduce the sample preparation conditions at the XFEL as closely as possible when performing the absorption/scattering measurement of the grease/buffer/crystal mixtures.

9) In a related issue, one reviewer finds it strange that the difference maps for the 10 ps data are much weaker compared to the ones at 1 ps despite better statistics in terms of multiplicity and to some extent also resolution based on the crystallographic Table 1. There is no reason given in the manuscript for the lower intensity of the difference map features. Is the initial photoactivated state just decaying that rapidly (possible but my guess is no) or were different pump-probe settings used for the 10 ps data? Please include any changes in the experimental settings between the two delays in the manuscript.

As the reviewers correctly state, the data statistics are better for 10ps compared to 1ps, but the difference signals are weaker at 10ps. This indicates that the population of the activated state is lower. We make this statement more clearly in the document. The lower population could be due to (a) decay of the activated states in the crystals and/or (b) annihilation between neighbouring activated states. While it would be interesting to find out the reason, we cannot provide a definite answer based on our data.

We estimated the noise levels of the difference maps by reducing the contour levels of our difference maps and by identifying the level at which spurious, noisy difference features appeared throughout the structure. We defined this as the background levels in our difference maps. The levels for 1ps are σ=3.0 (0.0095e/Å^3^) and for 10ps σ=3.4 (0.0094e/Å^3^) (Figure 1—figure supplement 4). Thus, the absolute electron density background level is lower for 10ps compared to 1ps, in agreement with the better statistics for 10ps compared to 1ps.

There were not any changes in the experimental settings, but a drift of the pump laser can never be fully excluded. This would lead to a change in spatial overlap between X-ray and optical laser and it would cause a difference in photoactivation yield.

We state now clearly that the photoactivation yield was lower at 10ps. We also state that the settings for 1ps and 10ps were nominally the same.

10) The 1 ps difference map is of high quality even though the authors state low quantum yields for the photoactivation of phytochromes in the order of 10%. Capturing low activation levels reliably is an important advantage of the technique and this study demonstrates this nicely. However, it would be interesting to have an estimate for the achieved levels of activation within the crystal to see how close they get to the stated quantum yields.

We estimated the quantum yield of photoactivation in the crystals to be 8%, corresponding to a factor α = 25, which was used to compute the extrapolated structure factor for the structure refinement. This should be considered to be the ratio between activated and non-activated phytochrome molecules, averaged over the entire crystal. We have now included information on how we determined the photoexcitation quantum yield in the supplementary information. The 8% corresponds to values which are typical in time-resolved spectroscopy experiments on femtosecond time scales.